# Enhancing CRISPR prime editing by reducing misfolded pegRNA interactions

Weiting Zhang[1,2], Karl Petri[3,4], Junyan Ma[1,2,5], Hyunho Lee[3,4], Chia-Lun Tsai[6], J Keith Joung[3,4], Jing-Ruey Joanna Yeh[1,2]*

[1]Cardiovascular Research Center, Massachusetts General Hospital, Charlestown, United States; [2]Department of Medicine, Harvard Medical School, Boston, United States; [3]Molecular Pathology Unit and Center for Cancer Research, Massachusetts General Hospital, Charlestown, United States; [4]Department of Pathology, Harvard Medical School, Charlestown, United States; [5]Medical College, Dalian University, Dalian, China; [6]Center for Computational and Integrative Biology, Massachusetts General Hospital, Boston, United States

**Abstract** CRISPR prime editing (**PE**) requires a Cas9 nickase-reverse transcriptase fusion protein (known as PE2) and a prime editing guide RNA (**pegRNA**), an extended version of a standard guide RNA (**gRNA**) that both specifies the intended target genomic sequence and encodes the desired genetic edit. Here, we show that sequence complementarity between the 5' and the 3' regions of a pegRNA can negatively impact its ability to complex with Cas9, thereby potentially reducing PE efficiency. We demonstrate this limitation can be overcome by a simple pegRNA refolding procedure, which improved ribonucleoprotein-mediated PE efficiencies in zebrafish embryos by up to nearly 25-fold. Further gains in PE efficiencies of as much as sixfold could also be achieved by introducing point mutations designed to disrupt internal interactions within the pegRNA. Our work defines simple strategies that can be implemented to improve the efficiency of PE.

*For correspondence:
jyeh1@mgh.harvard.edu

## eLife assessment

This **useful** paper reports on two simple methods for improving the efficiency of prime editing, a prominent gene editing technique. In combination with published modifications, the strategies described in this study may lead to significant improvements in editing efficiencies. The data are **solid**, and the methods will be of broad interest to anyone using gene editing.

## Introduction

PE is a versatile gene-editing technology that enables programmable installation of any nucleotide substitution and small insertions/deletions without requiring a DNA donor template or the introduction of DNA double-strand breaks (*Anzalone et al., 2020*). It utilizes a Cas9 nickase (nCas9)-reverse transcriptase (RT) fusion protein called PE2 and a prime editing guide RNA (pegRNA, *Figure 1a*). Like a gRNA, the pegRNA directs nCas9 to the target DNA site specified by the 5' spacer sequence. The non-target DNA strand nicked by nCas9 then anneals to a complementary primer binding site (PBS) at the 3' end of the pegRNA. Subsequently, the adjacent reverse transcriptase template (RTT) also encoded in the pegRNA is reverse transcribed by the PE2 RT generating a 3' DNA 'flap' that encodes the desired edit (*Anzalone et al., 2019*). Although PE has been successfully employed in mammalian cells, plants, *Drosophila*, zebrafish and mice, the editing efficiencies observed are generally lower than those observed with other forms of CRISPR/Cas-based editing (e.g. nucleases and base editors; *Anzalone et al., 2019*; *Bosch et al., 2021*; *Petri et al., 2022*; *Lin et al., 2020*; *Liu et al., 2020*).

To explore potential mechanisms for the lower editing frequencies observed with PE, we tested whether the 3' PBS/RTT segment might impact the ability of a pegRNA to function with standard SpCas9 nuclease. To do this, we compared the mutation frequencies induced by SpCas9 with matched pairs of pegRNAs and standard gRNAs (i.e. lacking the PBS and RTT 3' extensions) targeting the same spacer sequences. We performed these comparisons in one-cell stage zebrafish embryos by injecting Cas9 protein complexed with equimolar concentrations of either pegRNAs or gRNAs targeting six different endogenous gene loci. To quantify indel frequencies induced at these sites, we extracted genomic DNA from embryos one day following injection and performed targeted amplicon next-generation sequencing (NGS). We found that across all six target spacer sites, lower indel frequencies were observed with pegRNAs than with their matched gRNAs (*Figure 1b*), suggesting that the presence of the 3' extension in a pegRNA decreases Cas9-induced gene editing.

One potential explanation for the lower Cas9 activities we observed with pegRNAs compared with standard gRNAs is that interactions between complementary sequences in the 5' spacer and 3' PBS and RTT might cause misfolding of pegRNAs in a way that decreases their abilities to complex with Cas9 protein (*Figure 1c*).This possibility seemed likely given that previous work has shown that even shorter length internal interactions between bases in standard gRNAs can stabilize alternative non-functional folding and low SpCas9-induced mutation efficiencies (*Thyme et al., 2016*). To test our hypothesis, we used a previously described in vitro assay (*Thyme et al., 2016*) that assesses the abilities of various gRNAs (and, in this case, pegRNAs) to complex with Cas9 protein. In this assay, various matched pegRNAs and gRNAs targeted to endogenous zebrafish gene spacer targets compete with a gRNA targeting an *EGFP* reporter gene sequence for complexation with Cas9 nuclease (*Figure 1d*). The degree of successful competition by a given gRNA or pegRNA can be assessed by measuring cleavage of an *EGFP* target DNA site substrate included in each reaction (*Figure 1d*). Using this assay, we compared matched gRNAs and pegRNAs targeted to four different genomic DNA sites, *gpr78a*, *adgrf3b*, *cacng2b* and *gpr85* and found that in all four cases the gRNAs could efficiently compete with the *EGFP*-targeted gRNA for binding to Cas9 (as judged by reduced cleavage of the *EGFP* DNA target site template), whereas the pegRNAs were substantially and significantly reduced in this capability (*Figure 1e and f*; *Figure 1—source data 1 and 2*). To test whether this reduced Cas9-binding capability of the pegRNAs might be caused in part by their 5' vs 3' complementarity, we performed additional in vitro experiments with matched pegRNAs in which we introduced three point mutations in the 3' PBS region of the pegRNAs (*Figure 1—source data 3*) and found that these mutated pegRNAs showed significant increases in their competitive Cas9-binding activities (*Figure 1e and f*; *Figure 1—source data 1 and 2*).

Having determined that pegRNAs may suffer from reduced binding to Cas9 protein, we next considered whether refolding the pegRNAs could improve their Cas9 binding capabilities. We considered this hypothesis because it has been previously shown that refolding low-activity gRNAs believed to have alternative folding configurations can significantly improve their abilities to mediate Cas9-induced indels (*Thyme et al., 2016*). We found that subjecting pegRNAs to a refolding procedure consisting of heat denaturation followed by slow cooling (Materials and methods) significantly improved their abilities to bind Cas9 in the in vitro DNA cleavage competition assay (although not to levels observed with matched standard gRNAs *Figure 1e and f*; *Figure 1—source data 1 and 2*). Consistent with this, we also observed that re-folding of pegRNAs prior to formation and injection of pegRNA-Cas9 ribonucleoprotein complexes into zebrafish embryos also increased Cas9-mediated indel frequencies by up to 2.8-fold for four of the seven target sites tested (*Figure 1—figure supplement 1*).

Next, we evaluated whether refolding the pegRNAs would also increase PE efficiencies in zebrafish embryos. For this experiment we used 17 pegRNAs from our earlier study (*Petri et al., 2022*) that were designed to introduce various types of mutations (base substitutions, insertions, or deletions) and that had exhibited a range of PE efficiencies from barely active to the best performers. We found that refolding prior to embryo injection significantly increased the frequencies of pure PE (defined as the alleles containing only the pegRNA-specified edits without any other mutations) frequencies for nine of these 17 pegRNAs (*Figure 2a and b*). Significant increases were observed with seven of 12 pegRNAs designed to introduce base substitutions (increases ranging from 2.6- to 24.7-fold; *Figure 2a*) and 2 of 3 pegRNAs designed to create insertion pegRNAs (increases ranging from 1.7- to 4.6-fold; *Figure 2b*). We did not observe significant increases in PE efficiencies with the two pegRNAs

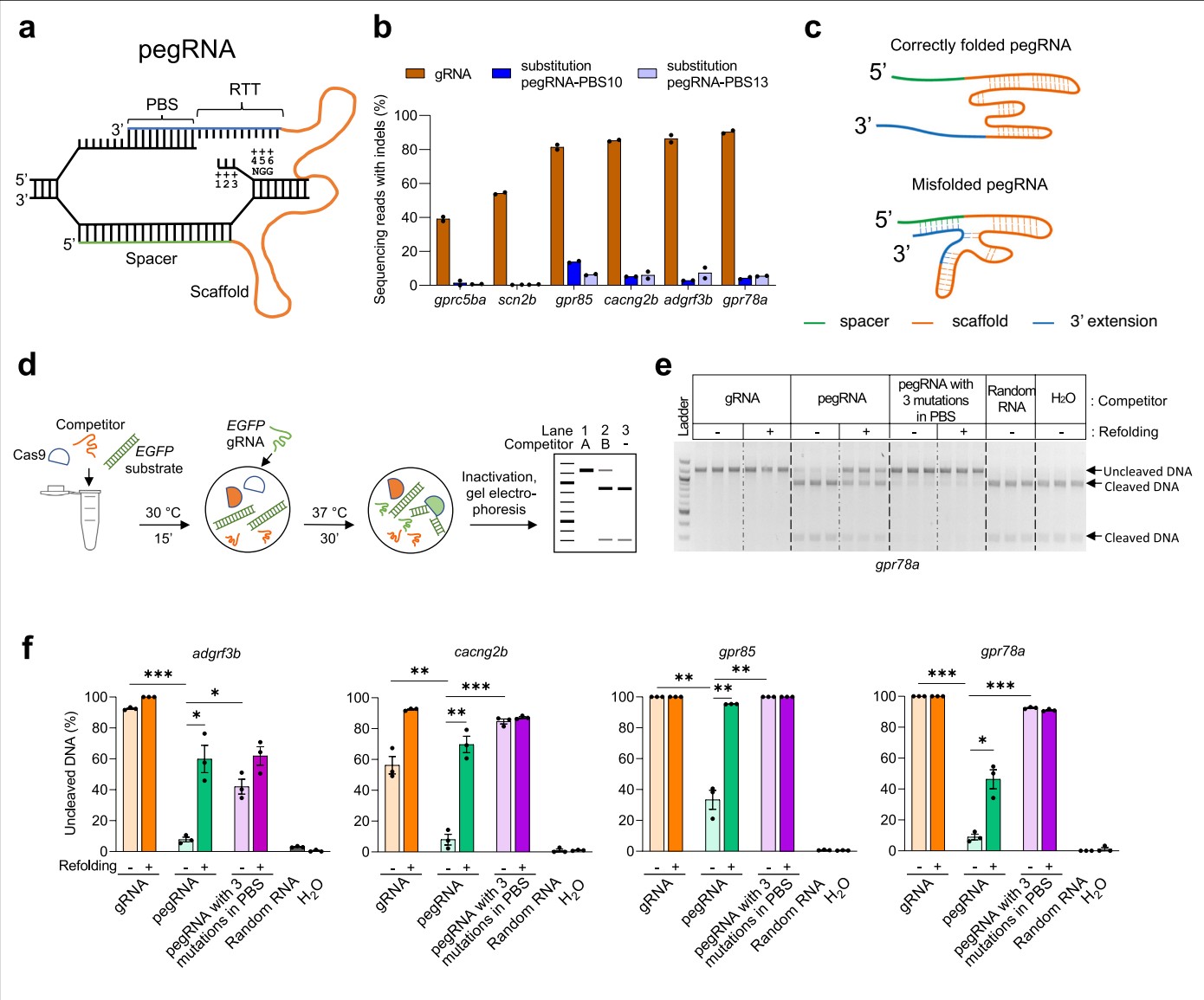

**Figure 1.** Improving in vitro SpCas9 binding efficiencies of pegRNA by refolding. (**a**) Schematic illustrating the hybridization of a pegRNA and its target DNA. Four segments of a pegRNA are shown. PBS, Primer Binding Site; RTT, Reverse Transcriptase Template. Target DNA positions (as well as the corresponding sequences in the RTT) are numbered counting from the SpCas9-induced nick towards 'NGG', the protospacer adjacent motif (PAM). (**b**) Mutation frequencies induced by SpCas9 with gRNAs and pegRNAs in zebrafish. All pegRNAs carried a single nucleotide substitution at position +5 or+6, with RTT lengths of 14- or 15-nucleotide (nt), and PBS lengths of 10-nt (pegRNA-PBS10) or 13-nt (pegRNA-PBS13). Target loci are indicated at the bottom. SpCas9 protein was complexed with gRNA or pegRNA at a molar ratio of 1:2 (0.6 µM of gRNA or pegRNA). (**c**) Schematic illustrating hypothetical conformations of correctly folded and misfolded pegRNAs. The spacer is shown in green, Cas9-binding scaffold in orange and 3' extension including PBS and RTT in blue. Dotted lines indicate potential base pairings. (**d**) Schematic illustrating the in vitro competition assay for Cas9 binding and substrate cleavage. Possible outcomes of the assay are shown in a representative gel. Lane 1 shows the addition of Competitor A with a high SpCas9-binding affinity resulting in 100% inhibition of cleavage of DNA substrates (1.2 kilobase pairs). Lane 2 shows the addition of Competitor B with a low SpCas9-binding affinity yielding a mix of uncleaved and cleaved (900 and 300 base pairs) DNA substrate. Lane 3 shows the reaction without any competitor resulting in 100% cleaved DNA products. (**e**) Agarose gel image showing the results of in vitro SpCas9 cleavage of DNA substrate in the presence of gRNA or pegRNAs targeting *gpr78a* as competitors, with or without refolding (indicated on top of the gel). Random RNA isolated from tolura yeast was used as a negative control. Assays were performed in triplicate. (**f**) Percentage of uncleaved DNA substrate in the presence or absence of competitor gRNA or pegRNA calculated using data from *Figure 1—source data 1* and *Figure 1—source data 2*. Competitor gRNA and pegRNA target loci are indicated at the top. Competitor types are shown at the bottom. Dots represent individual data points, bars the mean and error bars ± s.e.m. Unpaired two-tailed *t*-test with equal variance was used to compare non-refolded gRNA vs non-refolded pegRNA, non-refolded pegRNA vs non-refolded pegRNA with three mutations in PBS, and non-refolded vs refolded pegRNAs. *p<0.05, **p<0.01, ***p<0.001.

The online version of this article includes the following source data and figure supplement(s) for figure 1:

*Figure 1 continued on next page*

*Figure 1 continued*

**Source data 1.** Original agarose gel images showing in vitro SpCas9 cleavage of *EGFP* DNA substrate in the presence of gRNAs or pegRNAs targeting *gpr78* (**a**), *adgrf3b* (**b**), *cacng2b* (**c**), and *gpr85* (**d**).

**Source data 2.** Labeled agarose gel images showing in vitro SpCas9 cleavage of *EGFP* DNA substrate in the presence of gRNAs or pegRNAs targeting *adgrf3b* (**a**), *cacng2b* (**b**), and *gpr85* (**c**).

**Source data 3.** pegRNA with/without triple mutations in PBS for in vitro assay.

**Figure supplement 1.** Indel frequencies of non-refolded and refolded pegRNAs with SpCas9 in zebrafish.

designed to induce deletions (*Figure 2b*), perhaps due relatively lower degree of complementarity between the 5' spacer and 3' RTT because of the deletion encoded in the latter region. In general, increases in pure PE frequencies due to re-folding were accompanied by significant increases in non-pure PE frequencies (*Figure 2—figure supplement 1a, b*) and therefore PE purity values (defined as the percentage of pure PE edits out of total edits) did not show significant differences for most of the pegRNAs we assessed (*Figure 2—figure supplement 1c, d*). The only exceptions were two *scn2b* substitution pegRNAs that actually showed significantly improved PE purities with re-folding and the *cacng2b* deletion pegRNA that showed significantly reduced PE purity with refolding (*Figure 2—figure supplement 1c, d*).

Lastly, we tested whether introduction of single point mutations in the RTT region to reduce complementarity of pegRNA 5' and 3' regions might also lead to increases in PE efficiencies. Specifically, we explored the effects of creating mutations at the +1,+2, or +3 positions of the RTT (+1 defined as the first nucleotide just 5' to the first nucleotide of the PBS and +2 and+3 being further upstream within the pegRNA, *Figure 1a*) on PE efficiencies in zebrafish embryos. To do this, we introduced RTT + 1,+2, or +3 mutations into three pegRNAs that each mediated low PE frequencies even after re-folding (0.39–2.33%; *Figure 2c*) and that each encoded a single nucleotide substitution edit (encoded at RTT positions + 5 or+6). Following re-folding, complexation with PE2, and injection into zebrafish embryos, we found that pegRNAs harboring mutations at the RTT + 1 and+2 positions could make PE more efficient than their unmutated pegRNA counterparts (*Figure 2c* and *Figure 2—figure supplement 2a*). Based on these results, we introduced a mutation at the RTT + 2 position in five additional pegRNAs specifying a single edit and found that the mutation increased mean pure PE frequencies in all five cases (*Figure 2d*). Three of the five mutated pegRNAs showed statistically significant increases in pure PE frequencies and two of these three pegRNAs also showed significant increases in non-pure PE frequencies (*Figure 2d* and *Figure 2—figure supplement 2b*). Overall, a mutation at the RTT + 2 position increased pure PE frequency up to 6.7-fold (mean 2.4-fold; *Figure 2c and d*). PE purity was unchanged except for one pegRNA (*cacng2b*) that showed a modest but statistically significant increase (*Figure 2—figure supplement 2c*). Testing of three re-folded pegRNAs harboring the RTT + 2 mutations showed that they all induced higher indel frequencies with SpCas9 nuclease in zebrafish embryos than matched re-folded pegRNAs without the RTT + 2 mutations (*Figure 2—figure supplement 3*), suggesting that the +2 mutations function to enhance the abilities of these pegRNAs to form functional complexes with Cas9 protein.

## Discussion

Our work delineates and validates two simple and general strategies for improving the efficiency of PE that can be readily practiced by any investigator. Delivery of PE components as RNP complexes offers multiple potential advantages relative to DNA or RNA delivery (e.g. increased efficiency, reduced off-target effects, and avoiding risk of integration events; *Kim et al., 2014*; *Raguram et al., 2022*; *Burger et al., 2016*; *Kanchiswamy, 2016*; *Svitashev et al., 2016*; *Wang et al., 2020*; *Ponnienselvan et al., 2023*) and we show that combining pegRNA re-folding with the introduction of point mutations at the RTT +1 or+2 position can increase RNP-induced pure PE frequencies by as much as 29-fold in zebrafish embryos (*Figure 1*, *Figure 2* and *Supplementary file 1*). Our previous results showing that RNP-mediated PE functions in both zebrafish embryos and cultured human cells (*Petri et al., 2022*) suggest that these simple strategies should likely improve PE in other settings such as human and other cell types as well. The introduction of RTT mutations to reduce pegRNA 5' and 3' complementarity can also be used when practicing PE technology using non-RNP-delivery methods such as DNA or RNA transfection, transduction, and/or injection. Notably, *Li et al., 2022*

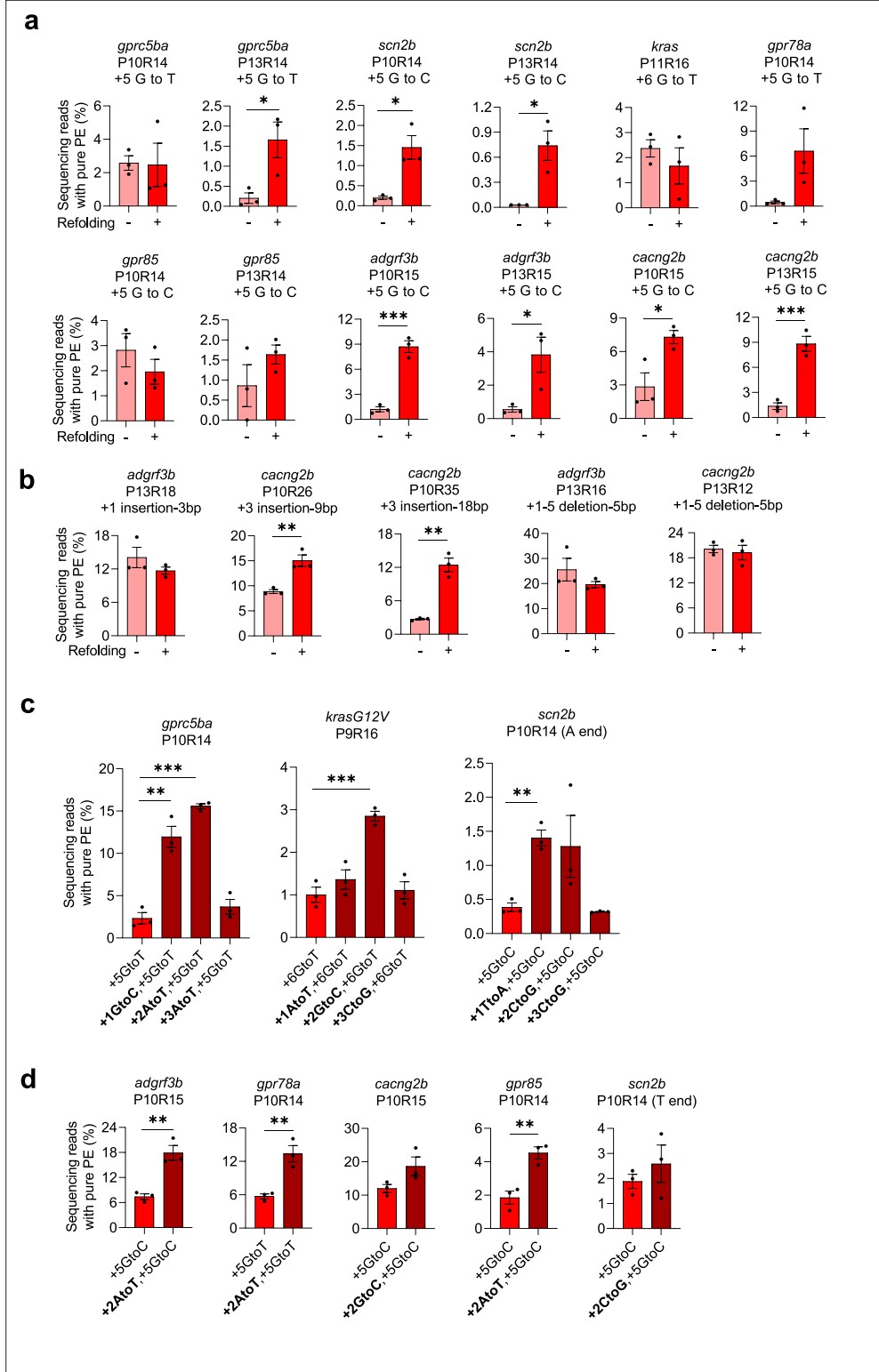

**Figure 2.** Improving prime editing efficiencies in zebrafish by pegRNA refolding and mutations in RTT. (**a–b**) Pure PE frequencies of non-refolded and refolded substitution pegRNAs (**a**) and insertion or deletion pegRNAs (**b**) with PE2 in zebrafish. Target loci, PBS lengths (labeled as 'P' followed by the number of nucleotides), RTT lengths (labeled as 'R' followed by the number of nucleotides), and pegRNA-specified edits (denoted as the position of the edit followed by the edit) are shown at the top. Pure PE represents sequencing reads containing only the pegRNA-specified mutations. (**c–d**) Pure PE frequencies with refolded pegRNAs carrying additional RTT mutations

*Figure 2 continued on next page*

*Figure 2 continued*

(at +1,+2 or+3) and PE2 in zebrafish. Target loci, PBS, and RTT lengths are shown at the top and pegRNA-specified edits are shown at the bottom. All pegRNAs had 3 or 4 thymine (T) nucleotides at the 3' end except for the ones labeled 'A end' for *scn2b* in which the terminal Ts were replaced with adenine (A) nucleotides. Dots represent individual data points (n=3 biologically independent replicates, 5–10 embryos per replicate), bars the mean and error bars ± s.e.m. *p<0.05, **p<0.01, ***p<0.001 (unpaired two-tailed *t*-test with equal variance).

The online version of this article includes the following figure supplement(s) for figure 2:

**Figure supplement 1.** Purities of prime editing with PE2 and non-refolded or refolded pegRNAs in zebrafish.

**Figure supplement 2.** Purities of prime editing with PE2 and refolded pegRNAs carrying additional RTT mutations in zebrafish.

**Figure supplement 3.** Indel frequencies in zebrafish induced by SpCas9 complexed with refolded pegRNAs with or without RTT mutation at +2 position.

---

have reported that introducing additional mutations in the RTT +1 to+3 positions can also enhance PE efficiency in human cells via DNA transfection. One important consideration in using the RTT mutation strategy is to ensure that the additional change introduced is either silent (if in a coding region) or otherwise benign in its effect (*Li et al., 2022*). Corroborating our findings, *Ponnienselvan et al., 2023* recently reported that 3' truncated pegRNAs are preferentially loaded onto Cas9 and the prime editor protein, potentially due to reduced pegRNA 5'–3' interactions. Although beyond the scope of this current work, it will be interesting to explore whether pegRNA re-folding and/ or RTT mutation might be combined with other previously described strategies for improving PE efficiencies (e.g. adding various structured RNA motifs to the 3' termini of pegRNAs *Nelson et al., 2022*; *Zhang et al., 2022* or PE proteins with improved architectures and activities *Nelson et al., 2022*; *Doman et al., 2022*; *Liu et al., 2021*).

Our findings also have more broad implications for both the design of pegRNAs and the range of genomic spacer sequences that can be targeted by PE for recognition and editing. Previous work has shown that the length and base composition of pegRNA PBS and RTT sequences can influence PE efficiency for any given target spacer site (*Ponnienselvan et al., 2023*; *Kim et al., 2021*; *Li et al., 2021*; *Lin et al., 2021*). Our work adds another parameter (internal complementarity between the 5' spacer and 3' PBS/RTT regions of the pegRNA) to be considered as one designs and tests various combinations to optimize PE activity. Accounting for this additional consideration may impact the nature of spacer sequences that can efficiently be targeted (e.g. higher GC content and/or melting temperature may actually be undesirable) and the length and composition of PBS/RTT sequences that can be used. We envision that the generation of larger datasets of optimized pegRNAs and consideration of all the parameters that can influence pegRNA activities (including internal complementarity) may yield improved rules and software for in silico design in the future.

## Materials and methods
### Production of gRNAs and pegRNAs

The gRNAs and pegRNAs (*Supplementary file 2*) used in this work were synthesized by in vitro transcription. The DNA templates for in vitro transcription were constructed by one-step PCR, using a C9E constant oligonucleotide containing the enhanced SpCas9 scaffold (*Petri et al., 2022*), forward primer carrying the SP6 promoter and target-specific spacer, and the reverse primer with the 3' extension containing the primer binding site and RTT. Primer sequences and PCR formulation are listed in *Supplementary file 3*. PCR reactions were conducted with Phusion High-Fidelity DNA polymerase (New England Biolabs, no. M0530L) using the following cycling program: 98 °C for 30 s, followed by 35 cycles of 98 °C for 10 s, 65 °C for 30 s, and 72 °C for 15 s, followed by a final 72 °C extension for 5 min. The PCR products were purified using the Monarch PCR & DNA Cleanup Kit (New England Biolabs, no. T1030L). In vitro transcription of pegRNAs and gRNAs was performed using the HiScribe SP6 RNA Synthesis Kit (New England Biolabs, no. E2070S), purified using the Monarch RNA Cleanup Kit (New England Biolabs, no. T2030L) and eluted in water.

## Purification of PE2 and SpCas9 protein

PE2 protein was purified as previously described (*Petri et al., 2022*). SpCas9 protein was purified as described in *Gagnon et al., 2014* with some modifications. Chemically competent Rosetta (DE3) competent cells (Novagen, no. 70954) were transformed with pET-28b-Cas9-His (Addgene, no. 47327) by heat shock following the manufacturers' instructions. 25 ml of overnight culture grown from a single colony in Luria Bertani (LB) medium with 50 µg ml$^{-1}$ kanamycin was transferred into 670 ml of autoinduction medium (24 g l$^{-1}$ yeast extract, 12 g l$^{-1}$ soy peptone, 12.5 g l$^{-1}$ potassium phosphate dibasic, 2.3 g l$^{-1}$ potassium phosphate monobasic, 0.4% glycerol) containing 50 µg ml$^{-1}$ kanamycin. The culture was incubated at 37 °C for about 4 hr until the OD$_{600}$ of the culture had reached 1.0–2.0 and was then switched to 18 °C for another 24 hr. Cells were collected and resuspended in lysis buffer consisting of 20 mM Tris-HCl pH 8.0, 20 mM imidazole, 150 mM NaCl, 0.1% Triton X-100, 5 mM 2-mercaptoethanol and cOmplete Protease Inhibitor Cocktail (Roche, no. 11697498001). The cell suspension was subjected to sonication (Qsonica) for 2 min (20 s pulses and 20 s rest between pulses) at 4 °C. Lysate was cleared by centrifugation at 14,000 rpm for 10 min at 4 °C. The supernatant was mixed with 2 ml Ni-NTA agarose (QIAGEN, no. 30250) and kept on a rotator at room temperature for 1 hr. Subsequently, the supernatant-agarose mixture was loaded onto an Econo-Column chromatography column (Bio-Rad, no. 7374011) and the supernatant flowed through by gravity. The flow-through was re-loaded onto the same column once more. The column was then washed with 80 ml of wash buffer (20 mM Tris pH 8, 20 mM Imidazole, 500 mM NaCl). Protein was eluted with 20 mM Tris pH 8, 250 mM Imidazole, and 300 mM NaCl, analyzed for purity by SDS–polyacrylamide gel electrophoresis (PAGE), and the elution buffer was replaced with 20 mM Tris pH 8.0, 200 mM KCl, 10 mM MgCl2, and 10% glycerol by dialysis using Slide-A-Lyzer G2 Dialysis Cassettes with 20 *kDa* cutoff (Thermo Fisher Scientific, no. 87737) at 4 °C overnight. Protein concentration was determined with UV absorbance at 280 nm on a NanoDrop spectrometer, and purified proteins were stored at −80 °C.

## Refolding of pegRNAs and RNP complexation of refolded pegRNAs with Cas9 or PE2

PegRNAs in water were refolded by heating at 98 °C for 2 min and slowly cooling down at a rate of –0.1 °C per second to 30 °C. Refolded pegRNAs were immediately added to the SpCas9 or PE2 protein, mixed gently, spun for 10 s at 6000 RPM in a mini centrifuge (Fisherbrand), and followed by a 10 min incubation at 30 °C.

## In vitro competition assay for Cas9 binding and substrate cleavage

We tested the ability of pegRNAs and matched gRNAs to inhibit Cas9-induced cleavage of a DNA substrate (*EGFP*) by competing with the cognate *EGFP* gRNA for binding to Cas9 (*Thyme et al., 2016*). For this assay 6 µl of Cas9 protein (16.67 ng/µl) in 400 mM KCl, 20 mM MgCl$_2$, and 40 mM Tris-HCL at pH 8.0. was mixed with 2 µl of gRNA or pegRNA being tested, or random RNA (SIGMA, no R6625) in a molar ratio of 1:2, and 2 µl of 25 ng/µl *EGFP* DNA substrate (1200 bp, amplified using the primers listed in *Supplementary file 4*). After incubating the mixture at 30 °C for 15 min, 2 µl of *EGFP* gRNA at the same molar concentration as the test gRNAs and pegRNAs, was added to the mixture and the cleavage reaction performed at 37 °C. After 30 min, the reactions were stopped by adding a gel loading dye (New England Biolabs, no. B7024A) followed by heat inactivation at 80 °C for 10 min. The cleaved DNA samples were separated on a 2% agarose gel by electrophoresis. The fluorescent intensity of each band was calculated by dividing its total fluorescent intensity measured in Image J/FIJI Gels by its band size in bp, yielding a unit fluorescent intensity for each band. The cleavage percentages were calculated by dividing the unit fluorescent intensity of the cleaved band at 900 bp by the sum of the values for the non-cleaved band at 1200 bp and the cleaved band at 900 bp.

## Zebrafish husbandry

All zebrafish (*Danio rerio*) husbandry and experiments were approved by the Massachusetts General Hospital Subcommittee on Research Animal Care and performed under the guidelines of the Institutional Animal Care and Use Committee at the Massachusetts General Hospital (Protocol #2005N000025).

## Zebrafish gene editing with Cas9 and PE2

Microinjections were performed using the one-cell stage of TuAB zebrafish embryos. Each embryo was injected with 2 nl of the RNP mixture at a Cas9 or PE2 protein to gRNA or pegRNA molar ratio of

1:2 (750 ng/µl of PE2 with 240 ng/µl of pegRNA, *Figure 2*; 500 ng/µl of Cas9 with 3 mixed pegRNAs of 80 ng/µl each, *Figure 1—figure supplement 1*, *Figure 2—figure supplement 3*) and immediately transferred to an incubator at 32 °C for PE2 and 28.5 °C for Cas9. For *Figure 1b*, six gRNAs, pegRNAs with a 10-nt PBS (pegRNA-PBS10), or pegRNAs with a 13-nt PBS (pegRNA-PBS13) were pooled together and mixed with Cas9 (20 ng/µl of each gRNA and 25 ng/µl of each pegRNA; Cas9 to gRNA/pegRNA molar ratio of 1:2) and injected as described above. One day post-fertilization, between five and ten embryos that developed normally from each condition were pooled and lysed in 10 mM Tris-HCl pH 8.0, 2 mM EDTA pH 8.0, 0.2% Triton X-100 and 100 µg ml$^{-1}$ Proteinase K (5–6 µl of lysis buffer/embryo). Lysates were incubated at 50 °C overnight with occasional mixing, heated at 95 °C for 10 min to inactive Proteinase K, and stored at 4 °C.

### Targeted deep sequencing

Amplicons for targeted sequencing were generated in two PCR steps. In the first step (PCR1), regions containing the target sites were amplified from 1 µl of the zebrafish embryo lysate, using touchdown PCR with Phusion High-Fidelity Polymerase (NEB, no. M0530S) and primers containing partial sequencing adapters (*Supplementary file 5*). For some samples, the products of PCR1 were purified using the Monarch PCR & DNA Cleanup Kit (New England Biolabs, no. T1030L) and deep sequenced at the MGH DNA Core. For the rest of the samples, the products of PCR1 were diluted 200-fold with water and used in the second PCR step (PCR2), where Illumina barcodes and P5/P7 sequences were attached to PCR1 products. The PCR2 product yield was assessed by agarose gel electrophoresis and pooled together in equal amounts. The PCR2 product pools were subjected to three steps of purification with the Monarch PCR & DNA Cleanup Kit (New England Biolabs, no. T1030L), Gel DNA Recover Kit (Zymoclean, no.11–300 C), and paramagnetic beads (1:1 beads/sample) using the same purification protocol as with AMPure XP beads (Beckman Coulter, no. B37419AB). The product purity was assessed via capillary electrophoresis on a QIAxcel instrument (Qiagen) and quantified by spectrophotometry (NanoDrop). The resulting sequencing libraries were sequenced using the MiSeq system (Illumina v.2 kit, 2×150 bp).

### Deep sequencing analysis

Sequencing data were analyzed with CRISPResso2 (*Clement et al., 2019*). Cas9 nuclease-treated samples were analyzed using Cas9 mode with a 5 bp quantification window size. PE-treated samples were analyzed using Prime editor mode with a 5 bp quantification window size and a 5 bp pegRNA extension quantification window size. CRISPResso2 was run with quality filtering (only those reads with an average quality score ≥30 were considered).

### Statistical analysis

For all bar graphs, mean and s.e.m. (only for samples with n>2) were calculated and plotted using GraphPad Prism v.8. Statistical analysis of the significance level was conducted using an unpaired two-tailed t-test with equal variance in Microsoft Excel version 2301.

## Acknowledgements

This work was supported by the Hassenfeld Scholar Award (to JRJY), NIH R01 GM134069 (to JRJY), and NIH RM1 HG009490 (to JKJ). KP was funded by the Deutsche Forschungsgemeinschaft (DFG, German Research Foundation) – Projektnummer 417577129. JM received support from the China Scholarship Council (CSC, 201808210354). We thank L Paul-Pottenplackel for help with revising the manuscript.

## Additional information

#### Competing interests

Karl Petri: has a financial interest in SeQure Dx, Inc. KP's interests and relationships have been disclosed to Massachusetts General Hospital and Mass General Brigham in accordance with their conflict of interest policies. J Keith Joung: has, or had during the course of this research, financial interests

in several companies developing gene editing technology: Beam Therapeutics, Blink Therapeutics, Chroma Medicine, Editas Medicine, EpiLogic Therapeutics, Excelsior Genomics, Hera Biolabs, Monitor Biotechnologies, Nvelop Therapeutics (f/k/a ETx, Inc), Pairwise Plants, Poseida Therapeutics, SeQure Dx, Inc, and Verve Therapeutics. JKJ's interests were reviewed and are managed by Massachusetts General Hospital and Mass General Brigham in accordance with their conflict of interest policies. JKJ is a co-inventor on various patents and patent applications that describe gene editing and epigenetic editing technologies. The other authors declare that no competing interests exist.

## Funding

| Funder | Grant reference number | Author |
| --- | --- | --- |
| National Institutes of Health | GM134069 | Weiting Zhang Jing-Ruey Joanna Yeh |
| National Institutes of Health | HG009490 | Karl Petri Hyunho Lee J Keith Joung |
| China Scholarship Council | 201808210354 | Junyan Ma |
| Deutsche Forschungsgemeinschaft | 417577129 | Karl Petri |
| Massachusetts General Hospital | Hassenfeld Scholar Award | Jing-Ruey Joanna Yeh |

The funders had no role in study design, data collection and interpretation, or the decision to submit the work for publication.

## Author contributions

Weiting Zhang, Conceptualization, Data curation, Formal analysis, Validation, Investigation, Visualization, Methodology, Writing – original draft, Writing – review and editing; Karl Petri, Hyunho Lee, Formal analysis, Investigation, Writing – review and editing; Junyan Ma, Resources, Writing – review and editing; Chia-Lun Tsai, Resources; J Keith Joung, Conceptualization, Resources, Supervision, Funding acquisition, Writing – original draft, Writing – review and editing; Jing-Ruey Joanna Yeh, Conceptualization, Resources, Supervision, Funding acquisition, Investigation, Methodology, Writing – original draft, Writing – review and editing

## Author ORCIDs

Jing-Ruey Joanna Yeh (iD) http://orcid.org/0000-0002-7768-4901

## Ethics

All zebrafish husbandry and experiments were approved by the Massachusetts General Hospital Subcommittee on Research Animal Care and performed under the guidelines of the Institutional Animal Care and Use Committee at the Massachusetts General Hospital. (Protocol #2005N000025).

Reviewer #1 (Public Review): https://doi.org/10.7554/eLife.90948.2.sa1
Reviewer #2 (Public Review): https://doi.org/10.7554/eLife.90948.2.sa2
Reviewer #3 (Public Review): https://doi.org/10.7554/eLife.90948.2.sa3

# Additional files

## Supplementary files

• Supplementary file 1. PE outcomes for pegRNAs with single and double substitutions and with or without refolding.

• Supplementary file 2. The target site, RT template, PBS and Cas9-binding scaffold sequences of pegRNAs.

• Supplementary file 3. PCR formulation and primers used in the DNA template construction for gRNAs and pegRNAs.

• Supplementary file 4. PCR primers for the generation of the EGFP DNA substrate.

- Supplementary file 5. PCR primers for deep sequencing.
- MDAR checklist

## Data availability

Deep sequencing data have been deposited in the NCBI Sequence Read Archive (project no: PRJNA995387).

The following dataset was generated:

| Author(s) | Year | Dataset title | Dataset URL | Database and Identifier |
|---|---|---|---|---|
| Zhang W, Petri K, Ma J, Lee H, Tsai CL, Joung JK, JRJ Yeh | 2023 | Enhancing CRISPR prime editing by reducing misfolded pegRNA interactions | https://www.ncbi.nlm.nih.gov/bioproject/?term=PRJNA995387 | NCBI BioProject, PRJNA995387 |

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
