## [Editor Report · eLife assessment]

This **useful** paper reports on two simple methods for improving the efficiency of prime editing, a prominent gene editing technique. In combination with published modifications, the strategies described in this study may lead to significant improvements in editing efficiencies. The data are **solid**, and the methods will be of broad interest to anyone using gene editing.

---

## [Referee Report · Reviewer #1 (Public Review)]

Summary: This work by Zhang et al. provides new strategies to improve the efficiency of precise Prime Editing (PE) in zebrafish embryos. The authors test how two simple changes impact PE efficiency: first, by refolding the pegRNA before complexing with Cas9 nickase-reverse transcriptase PE2, and second, by introducing mutations to the pegRNA intended to reduce its autoinhibitory activity by disrupting complementarity between the 5' spacer sequence and the 3' PBS-RTT (Primer Binding Site-Reverse Transcriptase Template).

Strengths: The authors tested multiple loci in the zebrafish genome to determine how pegRNA refolding and point mutations in the RTT would impact overall mutagenesis efficiency and precise PE at the target sites. The impact on efficiency was tested with three types of pegRNAs designed to introduce base substitutions, insertions or deletions. Next-generation sequencing of amplicons from pooled, injected embryos provided robust measurement of mutagenesis and editing. Insertion and deletion pegRNAs were overall more efficient than substitution pegRNAs, which may be useful information in considering experimental design strategy for introducing a specific variant. There is potential for further improvement by combining the authors' methods with previously published strategies to improve pegRNAs through design and chemical modification.

Weaknesses: The observed increases in the frequency of precise PE were relatively minor and inconsistent across the multiple pegRNAs tested. The substitution pegRNAs showed very low precise PE, at levels less than 1 percent, therefore the fold changes reported were still representative of 10 percent or less of overall edits. Overall mutagenesis frequency, as measured by indel formation, increased along with increased precise PE. The approach produces highly genetically mosaic embryos, therefore the utility for transient studies in injected zebrafish embryos is unclear. Data on improved germline transmission frequency of precise PE alleles would strengthen the study and be of wide interest in the zebrafish community.

---

## [Referee Report · Reviewer #2 (Public Review)]

Prime editing is a major gene editing technique because it allows for the introduction of all possible substitutions, as well as small insertions and deletions, without causing double strand breaks. However, its efficiency is often limited. In a previous study, the authors showed that prime editing could be performed in zebrafish using recombinant PE2 protein and pegRNAs generated by in vitro transcription, but at many of the sites tested, gene editing efficiency remained relatively low.

In this current paper, the authors find that when pegRNAs were combined with Cas9, many induced much less indels than their corresponding guide RNAs and propose that this is due to the complementarity between the 5' and 3' regions of pegRNAs. Two methods aiming to reduce the resulting circularization of pegRNAs were next shown to increase the efficiency of prime editing: a slow refolding protocol (which was previously shown to be useful for inefficient guide RNAs), and the introduction of a substitution at position +2 of the reverse transcriptase template sequence. The data obtained and analyzed is solid and convincing.

These methods are remarkably straightforward and proved beneficial for most of the pegRNAs tested. Consequently, they represent important advances for the prime editing technique.

It should be noted, however, that despite these advances, prime editing activity remained relatively low for a significant proportion of pegRNAs tested (with less than 2% sequencing reads exhibiting the expected sequence change). This shows that further improvements are still needed for this important gene editing technique.

---

## [Referee Report · Reviewer #3 (Public Review)]

In this study, Weiting Zhang et al., improved the editing efficiency of prime editor by reducing misfolded pegRNA interactions, and the improvement of efficiency for prime editor helped to expand its application range. It is a research paper on technology improvement. This study is somewhat innovative.